# Very early invasive angiography versus standard of care in higher-risk non-ST elevation myocardial infarction: study protocol for the prospective multicentre randomised controlled RAPID N-STEMI trial

Thomas A Kite [1], Amerjeet S Banning,[1] Andrew Ladwiniec [1], Chris P Gale,[2] John P Greenwood [1],[2] Miles Dalby,[3] Rachel Hobson,[4] Shaun Barber,[4] Emma Parker,[1] Colin Berry,[5] Marcus D Flather [1],[6] Nick Curzen [1],[7] Adrian P Banning,[8] Gerry P McCann,[1] Anthony H Gershlick[1]

For numbered affiliations see end of article.

**Correspondence to**
Dr Thomas A Kite;
tom.kite@nhs.net

## ABSTRACT

**Background** There are a paucity of randomised data on the optimal timing of invasive coronary angiography (ICA) in higher-risk patients with non-ST elevation myocardial infarction (N-STEMI). International guideline recommendations for early ICA are primarily based on retrospective subgroup analyses of neutral trials.

**Aims** The RAPID N-STEMI trial aims to determine whether very early percutaneous revascularisation improves clinical outcomes as compared with a standard of care strategy in higher-risk N-STEMI patients.

**Methods and analysis** RAPID N-STEMI is a prospective, multicentre, open-label, randomised-controlled, pragmatic strategy trial. Higher-risk N-STEMI patients, as defined by Global Registry of Acute Coronary Events 2.0 score ≥118, or >90 with at least one additional high-risk feature, were randomised to either: very early ICA±revascularisation or standard of care timing of ICA±revascularisation. The primary outcome is the proportion of participants with at least one of the following events (all-cause mortality, non-fatal myocardial infarction and hospital admission for heart failure) at 12 months. Key secondary outcomes include major bleeding and stroke. A hypothesis generating cardiac magnetic resonance (CMR) substudy will provide mechanistic data on infarct size, myocardial salvage and residual ischaemia post percutaneous coronary intervention. On 7 April 2021, the sponsor discontinued enrolment due to the impact of the COVID-19 pandemic and lower than expected event rates. 425 patients were enrolled, and 61 patients underwent CMR.

**Ethics and dissemination** The trial has been reviewed and approved by the East of England Cambridge East Research Ethics Committee (18/EE/0222). The study results will be submitted for publication within 6 months of completion.

**Trial registration number** NCT03707314; Pre-results.

## BACKGROUND AND RATIONALE

Historical data indicate that an invasive strategy during index hospital admission

### Strengths and limitations of this study

⇒ This randomised trial sought to test whether a very early invasive strategy in higher-risk non-ST elevation myocardial infarction patients improves clinical outcomes compared with standard care. An early invasive strategy in this group is recommended in international guidelines, but is as yet unsupported by the primary outcome of an appropriately sized randomised trial.

⇒ Randomised controlled pragmatic strategy design.

⇒ A cardiac MR substudy will provide mechanistic data on infarct size, myocardial salvage and residual ischaemia post percutaneous coronary intervention.

⇒ Due to the effects of the COVID-19 pandemic on clinical services and a lower than expected primary outcome event rate, trial recruitment was terminated early after enrolment of 425 patients (18.4% of intended).

in non-ST elevation myocardial infarction (N-STEMI) improves composite ischaemic outcomes, with maximal benefit seen in those at highest baseline risk for future major adverse cardiovascular events (MACE).[1] However, the optimal timing of invasive coronary angiography (ICA) and whether high-risk subgroups should be treated early remains controversial, despite it being a mandated management strategy in international guidelines.[2 3] Since N-STEMI is characterised by a risk-treatment paradox whereby higher-risk patients are less likely to receive aggressive pharmacotherapy and invasive management,[4] use and timing of ICA can differ significantly when compared with

Table 1 Higher-risk N-STEMI (GRACE score >140) subgroup analyses from randomised studies comparing early and delayed invasive strategies

| Trial | Patients | Time to ICA: early (median and IQR, hour) | Time to ICA: delayed (median and IQR, hour) | Primary outcome | Results |
|---|---|---|---|---|---|
| TIMACS 2009 | 961 | 14.0 (3.0–21.0) | 50.0 (41.0–81.0) | 6 months death, non-fatal MI, stroke | Early=13.9% Delayed=21.0% HR 0.65, 95% CI 0.48 to 0.89, p=0.006 |
| ELISA-3 2013 | 224 | 2.6 (1.2–6.2) | 54.9 (44.2–74.5) | 30-day death, non-fatal MI, recurrent ischaemia | Early=10.5% Delayed=19.1% HR 0.55, 95% CI 0.29 to 1.10, p=0.26 |
| RIDDLE-NSTEMI 2016 | 123 | 1.4 (1.0–2.2) | 61.0 (35.8–85.0) | 30-day death, non-fatal MI | Early=10.7% Delayed=17.9% HR 0.56, 95% CI 0.21 to 1.51 p=0.12 |
| VERDICT 2018 | 1025 | 4.7 (3.0–12.2) | 61.6 (39.4–87.8) | Death, non-fatal MI, refractory ischaemia, admission for heart failure at median 4.3 years | Early=34.0% Delayed=40.1% HR 0.81, 95% CI 0.66 to 0.99, p=0.023 |

GRACE, Global Registry of Acute Coronary Events; ICA, invasive coronary angiography; MI, myocardial infarction; N-STEMI, non-ST elevation myocardial infarction; TIMACS, Timing of Intervention in Acute Coronary Syndromes; VERDICT, Very Early Versus Deferred Invasive Evaluation Using Computerized Tomography.

the precisely defined management of STEMI. Clinically unstable patients require urgent revascularisation, while for others, the most appropriate timing of an invasive strategy is less certain.

A series of randomised trials have investigated this concept in unselected N-STEMI populations, thereby testing whether early revascularisation (<24 hours) improves clinical outcomes as compared with delayed or standard of care approaches.[5–18] Differences in study design, inclusion criteria, timing of ICA and endpoint definitions have resulted in conflicting results that are challenging to interpret and apply to current practice. When these data are evaluated in totality, patient-level meta-analysis has demonstrated no significant difference in death or myocardial infarction (MI) between the two strategies.[19]

The application of the Global Registry of Acute Coronary Events (GRACE) score in prior trial subgroup analyses has potential for risk stratification of those patients that may benefit most from expeditious revascularisation (table 1). A GRACE score >140 analysis of 961 patients from the Timing of Intevention in Acute Coronary Syndrome (TIMACS) trial showed that an early invasive strategy (14 hours post randomisation) reduced the risk of death, non-fatal MI and stroke at 6 months by 35% as compared with a delayed approach.[9] Moreover, the Very Early Versus Deferred Invasive Evaluation Using Computerized Tomography (VERDICT) study produced a similar finding, although in a subgroup of 1025 GRACE >140

N-STEMI patients who underwent a very early invasive strategy (4.7 hours post randomisation).[15] However, such analyses should only be considered hypothesis generating since: (1) the primary outcome in both overall trial populations was neutral and (2) the studies were undertaken in the era of conventional troponin and CK-MB, with up to one-quarter of patients exhibiting no biomarker rise.[9]

Given that currently available data are inconsistent and of insufficient scientific quality to inform best practice, a contemporary trial that prospectively investigates the timing of revascularisation in GRACE score defined high-risk N-STEMI is required to confirm or refute these prior observations.

## METHODS AND ANALYSIS
### Study design and inclusion criteria
The RAPID N-STEMI trial enrolled patients across 32 hospitals with on-site cardiac catheter laboratories in the UK. Potential participants who experienced symptoms within 12 hours prior to admission were assessed on attendance to hospital and the research team alerted if a diagnosis of N-STEMI was suspected. N-STEMI was defined as: (1) the presence of cardiovascular symptoms suggestive of myocardial ischaemia and (2) elevation in high-sensitivity troponin (hs-Tn) I or T. Risk stratification using the GRACE 2.0 score was then performed. Patients in whom the GRACE 2.0 score was ≥118, or ≥90 with at least one additional feature of high-risk presentation

**Table 2**  Rapid N-STEMI inclusion and exclusion criteria

| Inclusion criteria | Exclusion criteria |
|---|---|
| >18 years of age | ST elevation myocardial infarction |
| Clinical diagnosis of N-STEMI comprising:<br>► Cardiovascular symptoms suggestive of myocardial ischaemia<br>► Elevated high-sensitivity troponin I or T | Evident type 2 myocardial infarction |
| Symptoms<12 hours prior to admission | Previous known cardiomyopathy |
| GRACE 2.0 score ≥118 or if GRACE 2.0 score ≥90 but <118 must have at least one high-risk feature:<br>► Anterior location of ECG changes (V2–V5)<br>► ST segment depression in two contiguous leads of 0.15 mV/1.5 mm<br>► Diabetes mellitus on medication<br>► Elevated high-sensitivity troponin 3 × upper limit of normal | Need for urgent PCI according to ESC Guidelines (haemodynamic instability, VT, VF, recurrent or persistent pain) |
| Intention to perform angiography and, if indicated, follow-on revascularisation | Cardiogenic shock |
| Provision of verbal assent followed by written informed consent | Severe valvular heart disease |
| | Any contraindication to PCI |
| | Current participation in another intervention trial |

ESC, European Society of Cardiology; GRACE, Global Registry of Acute Coronary Events; N-STEMI, non-ST elevation myocardial infarction; PCI, percutaneous coronary intervention; VF, ventricular fibrillation; VT, ventricular tachycardia.

were deemed as higher risk. The full inclusion and exclusion criteria are listed in table 2.

Patients were enrolled after obtaining verbal consent once eligibility was confirmed in the emergency department or appropriate receiving unit. Participants were then randomised in a 1:1 fashion to either: group (A) very early ICA with a view to revascularisation or group (B) standard of care timing of ICA with a view to revascularisation. Research team members had 6 hours from hospital admission to randomise verbally consented patients who met all eligibility criteria (figure 1).

## Study procedures

Randomisation was performed via either a secure centralised web-based or telephone assisted system provided by http://www.sealedenvelope.com. Those assigned to very early angiography were transferred to the cardiac catheter laboratory as soon as possible. Teams were encouraged, but not mandated to achieve a randomisation to vascular sheath insertion time of less than 90 min. Timing of standard of care ICA was according to typical practice at individual UK centres but encouraged to be within 72 hours of admission. Percutaneous coronary intervention (PCI) and coronary artery bypass grafting (CABG) were performed according to current international guidelines.[20] Requirement for multivessel revascularisation to a non-infarct related artery was at the individual operator's discretion. Optimal medical therapy, including the use of antiplatelet agents, was in accordance with current clinical guidelines.[2] Drug-eluting stents were used in all cases unless there were clear contraindications. As this was a pragmatic strategy

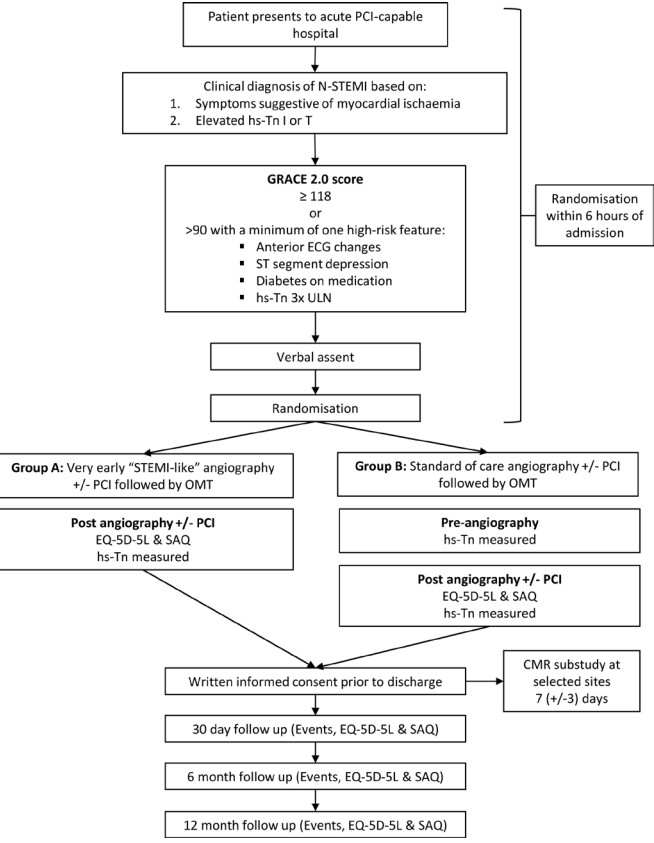

**Figure 1**  Rapid N-STEMI study flow diagram. CMR, cardiac MR; hs-Tn, high sensitivity troponin; N-STEMI, non-ST elevation myocardial infarction; OMT, optimal medical therapy; PCI, percutaneous coronary intervention; SAQ, Seattle Angina Questionnaire; STEMI, ST elevation myocardial infarction; ULN, upper limit of normal.

trial, all procedures performed were part of guideline directed standard of care for N-STEMI.

Patients were invited to provide verbal consent to participate in the study after being read an abridged consent form prior to randomisation. This was to avoid delay in those participants that were subsequently allocated to a very early invasive strategy. In addition to the baseline hs-Tn required for study inclusion, samples were obtained post angiography in both trial arms and prior to angiography in the standard of care arm. EQ-5D-5L[21] and Seattle Angina Questionnaires[22] to assess quality of life were completed after angiography in both arms. All patients were then asked to provide written informed consent for continuation in the study prior to hospital discharge. Study follow-up visits via telephone occurred at 30 days, 6 months and 12 months from randomisation. Clinical event reporting, EQ-5D-5L and Seattle Angina Questionnaire completion were performed at each of these time points.

### The GRACE 2.0 score

Previous high-risk subgroup analyses used the original GRACE score, with a score of >140 stratifying those patients at highest baseline risk. In the TIMACS and VERDICT trials, it was these groups that benefited from an early invasive strategy.[9 15] However, the updated GRACE 2.0 score demonstrates equivalent performance and is easier to implement in clinical practice as compared with the original GRACE score.[23] A notable advantage of GRACE 2.0 is that Killip Class and serum creatinine values are not required for risk calculation. This allows rapid stratification very early during hospital admission once a hs-Tn result is available, thus obviating the need to wait for renal biochemistry results.

Patients at intermediate risk (GRACE 2.0 score ≥90 to<118) were included to attenuate the perceived age bias of the GRACE score, thereby allowing enrolment of younger patients recognised to be at higher risk of future MACE. The following features: anterior ECG changes, ST segment depression, diabetes mellitus on medication and hs-Tn elevation three times the upper limit of normal, have been demonstrated as determinants of poorer prognosis in N-STEMI.[24–26]

### RAPID N-STEMI cardiac MR substudy

Imaging studies confined to N-STEMI are few and primarily descriptive, documenting smaller infarct size than in STEMI.[27] Novel cardiac magnetic resonance (CMR) markers (such as myocardial strain and salvage) may add incremental prognostic information to recognised predictors such as left ventricular ejection fraction (LVEF).[28 29]

The RAPID N-STEMI CMR substudy applied multiparametric CMR imaging to: (1) assess the impact of the timing of revascularisation on infarct size, volumes and LVEF, myocardial strain, myocardial salvage and extracellular volume and (2) quantify ischaemic burden post-PCI to ascertain whether this can predict risk of future MACE.

**Table 3** Rapid N-STEMI CMR study endpoints

| Primary outcome | Secondary outcomes |
|---|---|
| Infarct size (% left ventricular mass) | Left ventricular volumes and ejection fraction |
| | Myocardial salvage index |
| | Extracellular volume |
| | Ischaemic burden |
| | Global myocardial strain |

CMR, cardiac MR; N-STEMI, non-ST elevation myocardial infarction.

The RAPID N-STEMI CMR substudy endpoints are listed in table 3.

Four centres with an established record of high-quality CMR research participated in the substudy. As myocardial injury and infarct size reduces early following MI,[30] timing of CMR was standardised and performed at 7 (±3) days post admission. This also ensured angiography±PCI had been undertaken in both groups, as PCI itself may be associated with further myocardial injury.[31 32] The protocol included cine imaging in long and short axes. Adenosine stress perfusion was performed to assess for residual ischaemic burden and a gadolinium-based contrast agent administered to allow detection of myocardial necrosis and microvascular obstruction.[32] Where available, precontrast and postcontrast T1 mapping sequences were use to facilitate estimation of extracellular volume that may indicate more subtle changes in myocardial architecture.[33]

All CMR images will be sent to the core laboratory at the National Institute for Health Research (NIHR) Biomedical Research Centre in Leicester for quality control and central analysis, with the interpreting clinicians blinded to patient information and allocated group.

### Study endpoints

The RAPID-NSTEMI trial will evaluate the effect of a very early invasive strategy on the binary primary composite endpoint of all-cause mortality, non-fatal MI and admission for heart failure (HF) at 12 months following randomisation. The primary and secondary endpoints are listed in table 4.

### Sample size calculation

RAPID N-STEMI is a superiority trial powered to detect a 25% risk reduction in the primary endpoint. With a two-sided test of α=0.05% and 80% power, 964 patients were required in each arm of the study. Assuming up to 5% withdrawal, 5% crossover and 8% requiring CABG, 1157 patients were planned to be recruited to each group, resulting in a recruitment target of 2314.

Sample size calculations were based primarily on the subgroup analysis of GRACE>140 high-risk patients in the TIMACS study. The composite endpoint of death, non-fatal MI and stroke at 6 months occurred in 21.0% of patients in the standard care arm.[9] We decided to

**Table 4** Rapid N-STEMI study endpoints

| Primary outcome | Secondary outcomes |
|---|---|
| All-cause mortality, non-fatal myocardial infarction and admission for heart failure at 12 months | Individual components of primary composite outcome |
| | Cardiovascular mortality |
| | Ischaemia-driven revascularisation |
| | BARC 3–5 major bleeding |
| | Stroke |
| | Length of inpatient stay |
| | Admission for any cause |
| | Events prior to angiography |
| | Quality of life (Seattle Angina Questionnaire and EQ-5D-5L questionnaires) |
| | Cost–efficacy analysis |
| | Proportion of patients requiring emergency revascularisation in group B |
| | Total VARC-2 classified access site complications |
| | Major VARC-2 classified access site complications |

BARC, Bleeding Academic Research Consortium; N-STEMI, non-ST elevation myocardial infarction; VARC, Vascular Access Research Consortium.

include admission with HF since there is evidence of this being an important outcome following N-STEMI hospitalisation. Data from Kaul *et al* show that at 12 months following N-STEMI the incidence of admission with HF was 14.8%.[34] Based on these data and use of the GRACE 2.0 score, the standard care arm composite event rate of all-cause mortality, non-fatal MI and admission for HF in RAPID N-STEMI was estimated to be 19% at 12 months.

The CMR substudy is an exploratory mechanistic substudy that had a recruitment target of 200 patients. No formal power calculations were undertaken.

### Statistical analysis

A full statistical analysis plan will be completed prior to any analyses being undertaken. Primary analysis will be carried out by intention to treat with imputation for individuals with missing data due to lost to follow-up assessment as having no event. The primary outcome is binary for each individual (yes or no), and therefore, participants experiencing more than one of the composite events will only be counted once. The treatment arms will be compared using mixed effects logistic regression, which will be adjusted for randomisation stratification factors such as hospital site (as a random effect) and GRACE score (as a fixed effect). Treatment comparison estimates will be presented as adjusted ORs and 95% CIs.

The analysis of binary secondary outcomes will be carried out in the intention to treat population as per the primary outcome analysis. All other secondary continuous outcomes will be analysed on a complete case approach, where participants will only be included if relevant data are available.

Quality of life data (EQ-5D-5L) will be analysed using a mixed effects model with patient as a random effect to account for repeated measures over time. Each patient will contribute up to four postoperative repeated measures to the model. The model will be adjusted for the stratification variables as above. It is expected that some values at later time points will be missing. The mixed effects model specified here will include these patients with partially observed data.

### Subgroup analyses

An exploratory analysis of the primary outcome in line with the primary analysis plan will be repeated to investigate randomised treatment arm interaction with the following subgroups:

Gender: female and male.

Age at randomisation: <75 years and ≥75 years.

GRACE 2.0 score at admission: >140; >118 and <140; and 90–118.

ECG normal versus ECG abnormalities at admission.

### Exploratory analyses

The primary endpoint will also be analysed as a time-to-first-event outcome. The time will be measured from randomisation and differences between treatment arms compared using Cox's proportional hazards models, with treatment comparisons presented as hazard ratios and 95% CI. All time to event outcomes will be intention to treat with lost to follow-up censored at date last seen.

An exploratory analysis will be conducted repeating the analysis methods of the primary outcome in the efficacy population. The efficacy population excludes individuals that were randomised to early intervention not receiving angiography within 12 hours of randomisation OR were randomised to standard care receiving angiography within 12 hours (unless participant's procedure performed earlier than anticipated due to clinical deterioration).

The association between CMR outcomes and the primary outcome will be assessed by logistic regression with each CMR variable being included in a separate model. Models will have the clinical outcome as their dependant variable and include the CMR variable as explanatory variable as well as adjusting for treatment arm, site, GRACE score, age and sex.

### Ethics and dissemination

The study is conducted in accordance with the principles of the 1996 Helsinki Declarations, International Conference on Harmonisation-Good Clinical Practice (ICH-GCP) guidelines. The trial has been reviewed and approved by the East of England Cambridge East Research Ethics Committee (18/EE/0222). It is anticipated that data completion will be completed by the end

of December 2021, and the study results will be submitted for publication within 6 months of completion.

## Public and patient involvement

The study was presented to the NIHR University of Leicester Biomedical Research Centre (BRC) patient and public involvement group. Development of the protocol, outcome measures, recruitment to the trial and conduct of the study were discussed. There was a favourable response to the proposed study from the group. Study progress has been fed back to the patient and public involvement group during the course of the trial. Participants are given a study newsletter when they attend their 12-month clinic visit providing information about the study timelines and when the study results will be known. Access to the findings of the study will be made available in a contemporary and user-friendly way and full details of the results provided if the patient requests them.

## Trial coordination

Trial coordination is provided by the Leicester Clinical Trials Unit (LCTU) in collaboration with the chief investigator (CI) and the trial management group. LCTU is responsible for overall trial conduct including data management, quality assurance and statistical reporting. LCTU undertook site initiation visits, database training and ensures all aspects of the trial are performed to the highest ethical and research standards. The study is overseen by a Trial Steering Committee consisting of three experienced clinicians and the CI. An independent data and safety monitoring board convened to provide independent advice on study conduct and safety issues. Clinical events will be adjudicated by an independent Clinical Events Committee.

## Trial progress and impact of the COVID-19 pandemic

In March 2020, non-COVID-19 clinical research in the UK was suspended as NHS staff and resources were repurposed to frontline services in preparation for the volume of COVID-19 patients expected to place severe pressure on the NHS.[35] During this time, admissions with N-STEMI declined substantially.[36 37] Following the first wave in the UK, RAPID N-STEMI restarted in late July 2020 at a limited number of sites that had sufficient resource to recommence recruitment. However, due to the impending second wave of the COVID-19 pandemic the trial was once again suspended in December 2020. Discussions with the funding body took place regarding the strategy for a successful restart, with it agreed an interim pooled event rate (blinded to group allocation) should be calculated. Lower than anticipated event rates were documented and it was agreed with the funder that enrolment should be discontinued for two reasons: (1) the effect of the pandemic on clinical services and (2) the rate of the primary outcome. In summary, 425 (18.4% intended) patients were enrolled to the main trial, with 61 of these participants included in the CMR substudy. The intention is to perform in depth analyses of the

available data from these populations and present these in the near future.

## DISCUSSION

The optimal timing of revascularisation in higher-risk N-STEMI is a controversial topic, not least because international guidelines that mandate an early (<24 hours) invasive strategy are not supported by prospective randomised controlled clinical trial data.[2 3] RAPID N-STEMI addresses this knowledge gap. However, like many other clinical trials during the COVID-19 pandemic, RAPID N-STEMI was discontinued due to the emergency restructuring of healthcare and clinical research services. Despite falling short of the recruitment target, RAPID N-STEMI has randomised 425 GRACE score defined higher-risk patients admitted with N-STEMI, making it the third largest study to investigate this important patient population. It will therefore provide an significant contribution to the current evidence base, with dissemination of results planned for 2022.

### COVID-19: implications for cardiovascular research

The major challenges faced due to COVID-19 were threefold. First, significant reductions in admissions with acute coronary syndrome and HF occurred during the pandemic in the UK, with decreases of over to 40% in both disease entities, presumably due to fear of contagion in healthcare settings.[36 38 39] Not only did this reduce potential research participants, but such declines in admissions become a critical issue for clinical trial event reporting and thus may be a contributory factor to the lower event rates observed in RAPID N-STEMI.

Second, the NHS underwent the largest workforce redeployment since its inception to support severely pressurised frontline services treat the vast numbers of COVID-19 patients attending UK hospitals. Research staff were moved to such clinical areas, resulting in all non-COVID-19 related research being left severely disrupted and placed on hiatus until further notice,[40] the ramifications of which are sure to be felt long after the initial effects of the COVID-19 pandemic have abated.[41]

Third, and perhaps most fundamentally for RAPID N-STEMI, elective cardiology activity across the UK was effectively cancelled from the beginning of the first wave in March 2020.[35] Suspension of planned cases created greater catheter laboratory capacity for acute MI patients and dramatically reduced the standard of care timing to ICA for N-STEMI in the UK.[42] Such changes to catheter laboratory throughput and working patterns resulted in an unmanageable task of ensuring adequate time separation between the very early ICA and standard of care ICA arms in RAPID N-STEMI—essentially the control arm was accelerated. Since any potential differences in clinical outcomes are related to the difference in timing between the trial arms, new systems of care enforced by the COVID-19 pandemic left the RAPID N-STEMI

investigators in a position whereby restarting recruitment would inevitably jeopardise the scientific validity of the trial.

## Very early revascularisation in higher-risk N-STEMI: will optimal timing ever be defined?

RAPID N-STEMI is the fifth randomised trial to investigate the timing of an invasive strategy in GRACE score defined higher-risk N-STEMI patients, although the first to specifically investigate this higher-risk population. Prior studies report prespecified GRACE >140 subgroup analyses (table 1). Favourable results were observed in TIMACS and VERDICT,[9 15] while ELISA-3 and RIDDLE N-STEMI showed signals of benefit although were limited by small sample sizes.[12 14] Since clinical event rates are greater in high-risk N-STEMI, it may be expected that an adequately powered study to detect a clinically meaningful difference is achievable. Yet, in recent years outcomes following N-STEMI have improved, largely due to enhanced therapeutics and interventional techniques.[43] Therefore, demonstration of superiority for hard clinical endpoints from a very early invasive strategy may not be feasible in the contemporary era because the logistics of delivering an appropriately powered trial may be prohibitive.

The choice of trial endpoints is also of note. The recent Early or Delayed Revascularization for Intermediate and High-Risk Non ST-Elevation Acute Coronary Syndromes (EARLY) trial indicated benefit from a very early invasive approach in European Society of Cardiology defined high-risk patients (median GRACE score 122), but such benefit was driven by the softer endpoint of recurrent ischaemic events in a cohort that did not receive P2Y12 inhibitor pretreatment.[17] One may question the clinical relevance of an endpoint, and as such it was not included in the composite primary endpoint of RAPID N-STEMI. Given that practice in many centres is now shifting to a strategy of early ICA in higher-risk N-STEMI patient groups, and that this strategy is now widely accepted as without excess risk, it appears unlikely that the optimal timing of revascularisation in higher-risk N-STEMI will ever be robustly defined.

## Author affiliations
[1]Department of Cardiovascular Sciences, Glenfield Hospital, Leicester, UK
[2]Leeds Institute of Cardiovascular and Metabolic Medicine, University of Leeds and the Department of Cardiology Leeds Teaching Hospitals NHS Trust, Leeds, UK
[3]Royal Brompton & Harefield NHS Foundation Trust, London, UK
[4]Leicester Clinical Trials Unit, University of Leicester, Leicester, UK
[5]Institute of Cardiovascular and Medical Sciences, University of Glasgow, Glasgow, UK
[6]Norwich Medical School, University of East Anglia, Norwich, UK
[7]Faculty of Medicine, University of Southampton and University Hospital Southampton NHS Trust, Southampton, UK
[8]Oxford Heart Centre, Oxford University Hospitals NHS Foundation Trust, Oxford, UK

**Acknowledgements** The authors would like to acknowledge the support of the National Institute for Health Research Biomedical Research Centre in Leicester and the Leeds National Institute for Health Research Clinical Research Facility.

**Contributors** ASB, MDF, CB, NC, APB, GPM and AG conceived the idea for the study. ASB, GPM and AG designed the study protocol. TK drafted the manuscript. ASB, AL, MDF, CG, JPG, MD, RH, SB, EP, CB, NC, APB and GPM critically reviewed and approved the final version of the manuscript.

**Funding** The study is funded by the British Heart Foundation (grant number: CS/17/1/32445).

**Competing interests** CB is employed by the University of Glasgow which holds consultancy and research agreements for his work with Abbott Vascular, AstraZeneca, Boehringer Ingelheim, Causeway Therapeutics, Coroventis, Genentech, GSK, HeartFlow, Menarini, Neovasc, Siemens Healthcare and Valo Health.

**Patient and public involvement** Patients and/or the public were involved in the design, or conduct, or reporting, or dissemination plans of this research. Refer to the Methods section for further details.

**Patient consent for publication** Not applicable.

**Provenance and peer review** Not commissioned; externally peer reviewed.

## ORCID iDs
Thomas A Kite http://orcid.org/0000-0002-6021-5738
Andrew Ladwiniec http://orcid.org/0000-0002-1551-0594
John P Greenwood http://orcid.org/0000-0002-2861-0914
Marcus D Flather http://orcid.org/0000-0001-5644-3116
Nick Curzen http://orcid.org/0000-0001-9651-7829

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
