## [Reviewer comments · BMJ Open]

ARTICLE DETAILS

TITLE (PROVISIONAL)	Very early invasive angiography versus standard of care in higher-risk non-ST elevation myocardial infarction: study protocol for the prospective multicentre randomised controlled RAPID N-STEMI trial
AUTHORS	Kite, Thomas; Banning, Amerjeet S.; Ladwiniec, Andrew; Gale, Chris; Greenwood, John; Dalby, Miles; Hobson, Rachel; Barber, Shaun; Parker, Emma; Berry, Colin; Flather, Marcus; Curzen, Nick; Banning, AP; McCann, Gerry; GERSHLICK, ANTHONY

VERSION 1 – REVIEW

REVIEWER	Badings, Erik Deventer Ziekenhuis, cardiology
REVIEW RETURNED	17-Sep-2021

GENERAL COMMENTS	Relevant trial and well written manuscript. Unfortunately, the study was victim of the Covid 19 pandemic, as were many other trials. Although certainly true at the start of the study, I think the conclusion ("the study will provide further insights into higher risk N-STEMI....") is a bit too optimistic with knowledge of this premature abortion and in contradiction with the discussion section, stating that demonstration of superiority for hard clinical endpoints may not be feasible in the contemporary era. The authors might consider to rephrase this conclusion. Futhermore, I wonder why in this contemporary trial the latest universal definition of myocardial infarction is followed, in which not only an elevated Hs-Trop T of I is mandatory but also a rise and/or fall. However, I do realise that change of the inclusion criteria is not possible.
--

REVIEWER	Meah, Mohammed The University of Edinburgh, British Heart Foundation Centre for Cardiovascular Science
REVIEW RETURNED	03-Oct-2021

GENERAL COMMENTS	Would appreciate some further details in terms of the statistical analysis/endpoints: - Is the primary endpoint time to first event? Will repeat events and events of different types in the same person be reported? - Will the analysis be intention-to-treat? What is the plan for missing data? - Are any subgroup analyses planned on the primary endpoint outwith the CMR sub-study?
---

	Regardless of the above, this is an important study and will provide valuable insight despite limitations in its applicability due to the COVID-19 pandemic.
--	--

VERSION 1 – AUTHOR RESPONSE

Reviewer: 1

Dr. Erik Badings, Deventer Ziekenhuis

Comments to the Author:

1. Relevant trial and well written manuscript. Unfortunately, the study was victim of the Covid 19 pandemic, as were many other trials. Although certainly true at the start of the study, I think the conclusion ("the study will provide further insights into higher risk N-STEMI...") is a bit too optimistic with knowledge of this premature abortion and in contradiction with the discussion section, stating that demonstration of superiority for hard clinical endpoints may not be feasible in the contemporary era. The authors might consider to rephrase this conclusion.

Reply 1: Thank you for this comment. We agree and have removed this conclusions section of the manuscript.

2. Furthermore, I wonder why in this contemporary trial the latest universal definition of myocardial infarction is followed, in which not only an elevated Hs-Trop T of I is mandatory but also a rise and/or fall. However, I do realise that change of the inclusion criteria is not possible.

Reply 2: Thank you. We used a single elevated level of high-sensitivity troponin so that a very early strategy could be tested robustly, precluding the need to wait for a second sample taken at a later timepoint

Reviewer: 2

Dr. Mohammed Meah, The University of Edinburgh

Comments to the Author:

Would appreciate some further details in terms of the statistical analysis/endpoints:

1. Is the primary endpoint time to first event? Will repeat events and events of different types in the same person be reported?

2. Will the analysis be intention-to-treat? What is the plan for missing data?

3. Are any subgroup analyses planned on the primary endpoint outwith the CMR sub-study?

We would like to thank the reviewer for raising these important questions which was a significant oversight on our part. We have expanded the section in the manuscript describing the planned statistical analyses which now reads as below:

Statistical analysis

A full statistical analysis plan will be completed prior to any analyses being undertaken. Primary analysis will be carried out by intention to treat with imputation for individuals with missing data due to

loss to follow-up assessment as having no event. The primary outcome is binary for each individual (yes or no) and therefore participants experiencing more than one of the composite events will only be counted once. The treatment arms will be compared using mixed effects logistic regression, which will be adjusted for randomisation stratification factors hospital site (as a random effect) and GRACE score (as a fixed effect). Treatment comparison estimates will be presented as adjusted odds ratios (OR) and 95% confidence intervals (95% CI).

The analysis of binary secondary outcomes will be carried out in the intention to treat population as per the primary outcome analysis. All other secondary continuous outcomes will be analysed on a complete case approach, where participants will only be included if relevant data are available.

Quality of life data (EQ-5D) will be analysed using a mixed effects model with patient as a random effect to account for repeated measures over time. Each patient will contribute up to four postoperative repeated measures to the model. The model will be adjusted for the stratification variables as above. It is expected that some values at later time points will be missing. The mixed effects model specified here will include these patients with partially observed data.

Subgroup analyses

An exploratory analysis of the primary outcome in line with the primary analysis plan will be repeated looking for indications of a randomised treatment arm interaction with the following subgroups:

Gender; Female and Male

Age at randomisation: <75 years and ≥75 years

GRACE 2.0 score at admission; >140; >118 & <140; and 90-118

ECG normal vs ECG abnormalities at admission

Exploratory analyses

The primary endpoint will also be analysed as a time-to-first-event outcome. The time will be measured from randomisation and differences between treatment arms compared using Cox's proportional hazards models, with treatment comparisons presented as hazard ratios and 95% confidence interval. All time to event outcomes will be intention to treat with losses to follow-up censored at date last seen.

An exploratory analysis will be conducted repeating the analysis methods of the primary outcome in the efficacy population. The efficacy population excludes individuals that were randomised to Early Intervention not receiving angiography within 12 hours of randomisation OR were randomised to Standard Care receiving angiography within 12 hours (unless participant's procedure performed earlier than anticipated due to clinical deterioration).

The association between CMR outcomes and the primary outcome will be assessed by logistic regression with each CMR variable being included in a separate model. Models will have the clinical outcome as their dependant variable and include the CMR variable as explanatory variable as well as adjusting for treatment arm, site, GRACE score, age and sex.

Regardless of the above, this is an important study and will provide valuable insight despite limitations in its applicability due to the COVID-19 pandemic.

Thank you

VERSION 2 – REVIEW

REVIEWER	Meah, Mohammed The University of Edinburgh, British Heart Foundation Centre for Cardiovascular Science
REVIEW RETURNED	08-Jan-2022
GENERAL COMMENTS	The manuscript is much improved - many thanks for clarifying the plan with analyses. One small comment - I feel the subgroup analyses section should not be bullet pointed. Perhaps something like "...looking for indications of a randomised treatment interaction with subgroups including gender, age, grace score and ECG changes...". This is a stylistic comment and I respect the authors right to disagree.